# Do higher educated people feel happier?— Evidence of Han and minority nationalities from China

**Yuanyuan Wang[1], Zhenyi Hu[2], Jiameng Yuan[3], Kun Zhang[4]***

1 School of Accounting, Hebei University of Economics and Business, Shijiazhuang, Hebei, China, 2 School of Economics, Hebei University of Economics and Business, Shijiazhuang, Hebei, China, 3 Business School, The University of Sydney, Camperdown, New South Wales, Australia, 4 School of Economics, Guangdong University of Technology, Guangzhou, Guangdong, China

* kunzhang_uk@163.com

**Data Availability Statement:** Yes, all data is fully available and there are no restrictions. All data are from China General Social Survey (CGSS). The data is publicly stored in the database. Visit http://cgss.ruc.edu.cn/.

## Abstract

Based on the data of China General Social Survey (CGSS), this study adopts empirical analysis method to explore the impact of education on residents' subjective well-being and its differentiated mechanism in different ethnic groups. The results show that, first of all, education significantly improves residents' subjective well-being, and the conclusion is still robust after controlling for endogenous problems. Secondly, compared with Han nationality, education has a more significant effect on the subjective well-being of ethnic minority residents. Finally, by comparing the internal mechanism of education on the subjective well-being of Han and ethnic minority residents, the research finds that education mainly improves subjective well-being by improving residents' relative income level and enhancing their social class identification, in which the intermediary effect of income is particularly prominent. However, in the Han population, education may weaken the sense of fairness, and thus reduce happiness to some extent; This phenomenon has not been verified in ethnic minority groups. This study not only expands the literature on the relationship between education and subjective well-being, but also has important policy implications, providing a strong empirical basis for formulating more accurate education policies, improving the happiness of ethnic minority residents, and enhancing national cohesion.

## 1. Introduction

The pursuit of happiness is a universal aspiration, with subjective well-being serving as both a measure of individual life satisfaction and an essential indicator of societal harmony and stability. Enhancing subjective well-being has therefore become a focal point in academic discourse. Scholars have identified various macro- and micro-level determinants influencing well-being. At the macro level, factors such as economic development, public policy, employment, and environmental conditions play critical roles [1–5]. Micro-level influences include relative income, ethnicity and religion, consumption patterns, demographic characteristics such as age and gender, and health status [6–11].

**Funding:** The authors would like to acknowledge the financial support Funded by Humanities and social science fund of Ministry of Education of China [24YJC790231] awarded to KZ for this research. The funders provided invaluable assistance in the process of preparing the manuscript.

**Competing interests:** The authors have declared that no competing interests exist.

In recent years, scholars have increasingly focused on the relationship between education and subjective well-being, conducting in-depth investigations into its mechanisms. Education impacts subjective well-being through both direct and indirect channels. Directly, education enhances cognitive abilities and shapes values, significantly influencing life satisfaction in areas such as job satisfaction, interpersonal relationships, financial stability, and health [12, 13]. Highly educated individuals are more likely to experience positive emotions, such as happiness and contentment, while also mitigating negative emotions, such as depression and anxiety [14]. Indirectly, education improves subjective well-being by elevating relative income, enhancing social status, and strengthening communication skills [15–17].

Scholars have also explored the heterogeneity of education's impact on subjective well-being across various contexts. Cross-country analyses highlight significant variations in the mechanisms through which education influences well-being, shaped by differences in educational resource allocation, social security systems, and levels of economic development between developed and developing countries [18, 19]. Blanchflower (2021) demonstrated that, across both developed and developing countries, happiness generally follows a U-shaped curve with age, reaching its lowest point in middle age and gradually rising in later years [20]. Additionally, geographical disparities, particularly the urban-rural divide, further underscore the heterogeneity of education's impact. In urban areas, education is closely tied to career advancement and social status, whereas in rural areas, its impact is more pronounced in improving migration opportunities and family socioeconomic conditions [21]. Gender also moderates education's influence on subjective well-being. In many societies, education plays a more transformative role in improving women's happiness, especially in contexts where women historically had limited access to education [22].

Based on the above background, using data from the China General Social Survey (CGSS) for the years 2015, 2017, 2018, and 2021, this study empirically analyzes the impact of education on residents' subjective well-being and its differential effect among different ethnic groups. The results show that, first, education significantly improves residents' subjective well-being, and this conclusion remains robust after controlling for endogeneity. Second, compared with Han residents, education has a more significant effect on the subjective well-being of ethnic minority residents. Finally, through the comparison of the internal mechanisms by which education affects the subjective well-being of Han and minority residents, the research finds that education mainly enhances well-being by improving residents' relative income levels and strengthening their sense of social class identity, with income serving as a particularly prominent intermediary. However, among the Han population, education may weaken their sense of fairness, thereby reducing happiness to some extent; this phenomenon has not been observed among ethnic minority groups.

This paper contributes to the existing literature in several significant ways. First, it incorporates education and residents' subjective well-being into a unified research framework, utilizing relevant data from four periods (2015, 2017, 2018, and 2021) to examine the overall impact of education on Chinese residents' well-being since the country entered the deepening period of reform.

Second, although the existing literature has extensively explored the differential effects of education on subjective well-being across various groups, it has primarily focused on dimensions such as macro-cultural contexts [18, 19], individual age [20], gender [22], and geographical location [21]. However, previous studies have often overlooked the important role that ethnic factors play in influencing the relationship between education and residents' subjective well-being. This paper addresses this gap by examining the impact of education on the subjective well-being of both Han and ethnic minority groups, incorporating ethnic factors into the analytical framework. This approach provides new theoretical and empirical support for

understanding the role of education in improving well-being in ethnic minority areas, while also enriching the literature in the field of ethnic studies.

Third, through a comparative analysis of the mechanisms by which education impacts the subjective well-being of Han and minority ethnic groups, this paper explores how education influences these groups through three key channels: relative income, social class positioning, and perceptions of social fairness. This comprehensive exploration provides a more nuanced understanding of the differential effects of education, thereby enhancing the practical applicability of the findings to policy interventions aimed at improving the well-being of diverse ethnic populations.

Finally, this paper rigorously examines the multi-faceted role of education in enhancing the subjective well-being of both Han and minority residents, offering empirical evidence of the positive impact of government and Party education policies on the welfare of minority groups. These findings contribute to the broader discussion surrounding the role of education in socioeconomic development, providing important theoretical and practical implications for policymakers working to improve well-being and promote social equity among different ethnic groups.

## 2. Theoretical analysis and research hypothesis

In recent years, scholars increasingly focused on the impact of education on residents' subjective well-being and conducted in-depth research on the topic. Based on self-determination theory, Deci and Ryan (2020) pointed out that education enhanced individuals' autonomy, sense of competence, and sense of belonging, thereby improving intrinsic motivation and ultimately boosting subjective well-being [23]. Social comparison theory provided another perspective for analyzing the mechanisms through which education affected happiness. According to this theory, people assessed their happiness by comparing their social status and achievements [24]. Education elevated an individual's position in social comparisons, thereby enhancing their happiness. The human capital model also explained the impact of education on subjective well-being. By improving skills and productivity, education increased individuals' labor market returns and raised their standard of living. Human capital was defined as the "knowledge, information, personal ideas, and skills" possessed by individuals [25], with education being the most critical investment in human capital [26]. Individuals with higher-quality education typically acquired more advanced knowledge and abilities, enhancing their competitiveness in the job market [27].

Research on the mechanisms through which education influences residents' subjective well-being primarily focused on its direct and indirect effects. The direct effect highlights that education not only enhances cognitive abilities but also significantly improves individuals' subjective assessments of their well-being [28]. This is reflected in how education shapes subjective evaluations of life quality by influencing values, expectations, historical experiences, and future outlooks, including satisfaction with job roles, relationships, finances, and health [12, 13]. From an emotional perspective, education impacts individuals' emotional experiences by affecting the frequency and intensity of positive and negative emotions. Positive emotions include happiness, excitement, and satisfaction, while negative emotions include sadness, anger, and frustration. Ross and Van Willigen (1997), analyzing American family data, found that individuals with higher education levels were less likely to experience depression, anxiety, and anger compared to those with lower education levels [14]. The mediating effects of education emphasize its influence on factors such as income, social communication skills, class status, and physical and mental health, all of which affect residents' subjective well-being [15–17, 29].

However, some scholars have questioned the universally positive impact of education on happiness, arguing that issues such as educational inequality, mismatch, heightened expectations, and related stress can diminish its benefits [30]. Different educational levels lead to varying expectations regarding returns on education at different stages of development [13, 31]. Nikolaev (2018) found that individuals with higher education levels often have elevated expectations for future success, which can reduce subjective well-being if these expectations are unmet [32]. Moreover, as happiness is a subjective evaluation, it is influenced by social comparisons and reference points. Education broadens perspectives and shifts reference points beyond local or specific groups. These new reference points, while critical in determining happiness, can diminish subjective well-being when individuals perceive disparities in living conditions. Happiness is formed through the interaction between individuals and society and becomes gradually balanced and stable over time. Improving residents' educational levels can help individuals enhance their skills, create human capital advantages, and improve subjective well-being. Additionally, education provides certain non-monetary benefits, having a positive effect on individual subjective well-being. Based on this, this paper proposes Research Hypothesis 1:

**H1: Improving residents' educational level helps to improve residents' subjective well-being.**

Although extensive research has explored the impact of education on residents' subjective well-being, the existing literature often neglects the mechanisms underlying the differential effects of education on the subjective well-being of residents from different ethnic groups (Han majority and ethnic minorities). Differences in values and cultural backgrounds among ethnic groups represent a crucial factor shaping this relationship. Studies have identified significant disparities between Han and ethnic minority groups in terms of cultural context, social structures, and value orientations [33]. Fischer and Karl (2022), in their review of cross-cultural studies, emphasized that variations in value systems across cultural contexts significantly influence individuals' perceptions and evaluations of educational benefits [34]. Kim et al. (2015) demonstrated that ethnic groups differ markedly in social norms and networks, and these cultural differences may constrain the role of education in fostering social capital accumulation among ethnic minorities, thereby limiting its contribution to their subjective well-being [35]. Cheng (2021) further revealed that ethnic minority groups in China face substantial disparities in economic returns to education, which are closely tied to the structural barriers they encounter in the labor market [36].

Ethnic culture and religious beliefs are critical factors influencing subjective well-being [7, 37]. National identity and pride significantly shape individuals' thoughts, emotions, and behaviors, and are positively associated with general well-being [37, 38]. However, the dominant role of majority ethnic groups in cultural and social values, coupled with their advantages in resource allocation, often marginalizes ethnic minorities, hindering their integration into mainstream culture and resulting in lower levels of subjective well-being [39]. Stevenson and Wolfers (2012) observed that, despite the narrowing of the Black-White happiness gap between the 1970s and 2000s, substantial disparities persist [40]. Similarly, research from Iceland (2019) indicates that racial and ethnic inequalities remain widespread in the U.S., affecting income, education, health, living conditions, and career opportunities [41]. Assari (2018) further highlighted that upward social mobility and socioeconomic resources gained through education have a less pronounced positive impact on the health and well-being of African Americans and other minorities compared to Whites [42].

Enhancing income levels and educational attainment, while reducing inequalities, is considered an effective strategy for improving quality of life and general well-being. In China,

regional educational inequality has long been a structural issue within the higher education system. The eastern regions exhibit significantly higher levels of higher education development compared to the central and western regions, where ethnic minorities are predominantly concentrated. To address this imbalance, the Chinese government has implemented systematic measures to improve higher education across regions and enhance national educational capacity. Raising the educational attainment of ethnic minority residents not only safeguards their right to education but also elevates the development quality of minority-dominated areas. This, in turn, helps narrow the gap in livelihood development between ethnic minority and Han regions, positively influencing subjective well-being.

Moreover, improving education enables ethnic minority residents to break free from fixed ethnic identities, entrenched social roles, and class stereotypes. It eliminates ethnic barriers, fosters cultural integration, enhances the psychological sense of belonging, and supports the formation of a nested social structure that links ethnic minorities with the Han majority. By ensuring equitable access to livelihood services, development opportunities, and public welfare, education can significantly enhance the subjective well-being of ethnic minority residents.

**H2: Ethnic factors have a positive moderating effect on the level of education and residents' subjective well-being.**

## 3. Data source, model setting and variables

### 3.1 Sample selection

The Chinese General Social Survey (CGSS) is a national, comprehensive, and continuous survey project initiated by the Chinese Academy of Social Sciences (CASS) and implemented by the National Survey Research Center at Renmin University of China (NSRC). The CGSS employs a multi-stage, stratified, and probabilistic sampling method designed to capture a nationally representative sample of the Chinese adult population. This sampling process involves selecting respondents from various provinces, cities, and rural areas, ensuring that the sample accurately reflects China's geographic, demographic, and socio-economic diversity. The survey adheres to the ethical review standards established by the Chinese Academy of Social Sciences. This study utilized CGSS data from 2015, 2017, 2018, and 2021 to construct a mixed panel dataset. During data processing, samples of non-ethnic minorities with missing values in key variables (such as inability to answer or refusal to answer) were excluded, resulting in a final dataset of 40,431 observations. To minimize the influence of extreme values, continuous variables were winsorized at the 1% and 99% quantiles.

### 3.2 Model setting

Given that the two main variables, educational level and residents' subjective well-being, are ordinal in nature and possess an inherent hierarchical structure, the ordered logit model is particularly suitable for this analysis. This model effectively captures the ordered characteristics of the data while providing robust estimates of how independent variables influence the likelihood of different outcome levels. Accordingly, this study employs the ordered logit model as the primary analytical method. To verify Hypothesis 1: Improving residents' educational level helps to improve residents' subjective well-being. This paper sets the benchmark model as follows:

$$happiness_{i,t}^{*} = \beta_0 + \beta_1 eduhigh_{i,t} + \beta_2 Controls_{i,t} + \sum YearFE + \varepsilon_{i,t} \tag{1}$$

$$
\text{happiness}_{i,t} = 
\begin{cases}
1, & \text{if happiness}_{i,t}^* \leq \beta_1 \\
2, & \text{if } \beta_1 < \text{happiness}_{i,t}^* \leq \beta_2 \\
3, & \text{if } \beta_2 < \text{happiness}_{i,t}^* \leq \beta_3 \\
4, & \text{if } \beta_3 < \text{happiness}_{i,t}^* \leq \beta_4 \\
5, & \text{if happiness}_{i,t}^* > \beta_4
\end{cases}
\tag{2}
$$

The $\text{happiness}_{i,t}^*$ represents the subjective well-being of the i resident in t year. $\beta_1 < \beta_2 < \beta_3 < \beta_4$ are all parameters to be estimated. $eduhigh_{i,t}$ represents the educational attainment of the ith resident in year t. $\varepsilon_i$ is the random disturbance term.

To verify Hypothesis 2: ethnic factors have a positive moderating effect on education level and residents' subjective well-being. This paper sets the model as follows:

$$
\begin{aligned}
happiness_{i,t}^* &= \beta_0 + \beta_1 edu_{i,t} + nation_{i,t} + nation_{i,t} * eduhigh_{i,t} \\
&\quad + \beta_2 Controls_{i,t} + \sum YearFE + \varepsilon_{i,t}
\end{aligned}
\tag{3}
$$

The $nation_{i,t}^*$ represents the ethnic attribute of the ith resident in year t.

## 3.3 Varibles

**3.3.1 Residents' subjective well-being (happiness).** Residents' subjective well-being (**happiness**) is the core explained variable in this paper. This paper uses the CGSS questionnaire "In general, do you think your life is happy?" As an indicator of subjective well-being, the specific options of the question include: 1 = very unhappy, 2 = relatively unhappy, 3 = not happy or unhappy, 4 = relatively happy, and 5 = very happy [43]. Although subjective well-being is a subjective evaluation, and there is still a certain deviation to reflect the objective reality such as the current situation of life, such evaluation is also made based on certain objective conditions, which can reflect residents' perception of life to some extent. Therefore, it is still a scientific practice to use subjective well-being as the proxy variable of happiness, and this kind of question is often used as a measurement index in existing studies.

**3.3.2 Educational level (eduhigh).** Educational level (**eduhigh**) is the core explanatory variable in this paper. In the CGSS questionnaire, participants were divided into 13 levels of education from below primary school to above master's degree. The details are: 1 = no education, 2 = private school, literacy class, 3 = primary school, 4 = junior high school, 5 = vocational high school, 6 = general high school, 7 = technical secondary school, 8 = technical school, 9 = junior college (adult higher education), 10 = junior college (formal higher education), 11 = bachelor's degree (adult higher education), 12 = bachelor's degree (formal higher education), 13 = Graduate and above. Based on the actual situation of China's education system and experience with the questionnaire responses, this paper has processed the data to some extent. Individuals without any education were assigned a level of 0. The educational level of private schools, literacy classes and primary schools was assigned a value of 1, junior high schools were assigned a value of 2. The vocational high schools, general high schools, technical schools and technical schools were assigned a value of 3. The college junior college (adult higher education, formal higher education) was assigned a value of 4. The undergraduate college (adult higher education) and undergraduate college (formal higher education) were assigned a value of 5, and graduate students and above were assigned a value of 6.

**3.3.3 Mediating variables.** Relative income level (**economy**) is one of the mediating variables in this paper. This paper will use the question "Which level does your family's economic

**Table 1. Definition and description of research variables.**

| Variable type | Variable symbol | Variable name | Variable definitions |
|---|---|---|---|
| Explained variable | happiness | Residents' well-being | See the previous section for detailed measurement methods of indicators |
| Core explanatory variable | eduhigh | degree of education | See the previous section for detailed measurement methods of indicators |
| Moderator | nation | nation | 1 for ethnic minorities and 0 for Han |
| Mediating variable | economy | relative income | See the previous section for detailed measurement methods of indicators |
| | status | Class location | See the previous section for detailed measurement methods of indicators |
| | fairness | Sense of fairness | See the previous section for detailed measurement methods of indicators |
| Control variable | status14 | Class location at 14 years old | Same as status |
| | gender | gender | It is 1 for men and 0 for women |
| | party | Whether a member of the Communist Party of China | Communist party is 1, the other is 0 |
| | child | Number of children | Number of children |
| | marriage | Marital status | Married to 1, the other is 0 |

status belong to in your location?" in the CGSS questionnaire [44]. The highest "5" represents well above average and the lowest "1" represents well below average.

**Status** is the mediating variable in this paper. This paper will use the question "In our society, some people are at the upper and some at the lower levels" in the questionnaire [43]. The highest '10' represents the top level, and the lowest '1' represents the bottom level." At the same time, this paper combines the ratings of the strata in the way of combining the two levels into one, and the last 10 levels into five levels. Among them, "5" represents the top layer and "1" represents the bottom layer.

The **fairness** is also one of the mediating variables in this paper. This paper will use the question "In general, do you think today's society is fair or not?" in the questionnaire. The lowest "1" represents complete unfairness and the highest "5" represents complete fairness. The higher the value of this variable, the more equitable the society is.

**3.3.4 Control variable.** In this paper, the following variables are selected to be controlled: gender, number of children, marital status, and year. The specific definitions and descriptive statistics of variables are shown in Table 1.

## 3.4 Descriptive statistics and results

Table 2 provides details of the variables' descriptive statistics. We can see that the average happiness is 3.886, the maximum value is 5, and the minimum 1. The average value of eduhigh is 2.189, the standard deviation is 1.502, the minimum value is 0, and the maximum value is 6. The average value of happiness and eduhigh is basically consistent with previous research results. This paper calculates the variance inflation factor (VIF) of the model in the regression analysis, and the results show that the inflation factor of this study is far less than 10, which proves that there is no serious multicollinearity problem in this paper.

We also compare the educational level of ethnic minority residents and Han residents with the data of residents' subjective well-being, as shown in Table 3. The sample data show that among ethnic minority residents, the education level of primary school or below is 11% higher than that of Han residents; In the proportion of junior middle school education, minority residents and Han residents are basically the same; In the proportion of senior high school

**Table 2. Descriptive statistics of main research variables.**

| Variable | Average value | Standard deviation | Minimum value | Maximum value |
|---|---|---|---|---|
| happiness | 3.886 | 0.828 | 1 | 5 |
| eduhigh | 2.189 | 1.502 | 0 | 6 |
| nation | 0.0751 | 0.264 | 0 | 1 |
| economy | 2.551 | 0.765 | 1 | 5 |
| status | 2.420 | 0.862 | 1 | 5 |
| fairness | 3.203 | 1.027 | 1 | 5 |
| status14 | 1.912 | 0.916 | 1 | 5 |
| gender | 0.468 | 0.499 | 0 | 1 |
| party | 0.110 | 0.313 | 0 | 1 |
| child | 1.687 | 1.225 | 0 | 6 |
| marriage | 0.767 | 0.423 | 0 | 1 |

education, junior college education and above, the proportion of ethnic minority residents is basically lower than that of Han residents. This shows that compared with Han residents, the overall educational level of ethnic minority residents is still lower, and the proportion of low-educated groups in the total population is relatively high. To a certain extent, this shows that although the Chinese government's preferential education policies for ethnic minorities have played a great role in significantly improving the educational level of ethnic minority residents, there is still a certain gap between ethnic minority residents and Han residents in terms of education.

According to the data of residents' subjective well-being, there is no obvious difference between ethnic minority and Han residents. The data initially show that as a multi-ethnic country, the happiness of people of all ethnic groups is relatively high, which in a sense also indicates that the Chinese government's ethnic policy of ethnic equality and common prosperity has been successfully implemented and achieved good results.

**Table 3. Comparison of educational level and residents' happiness data between Han and minority nationalities.**

| | | Han | | | Minority nationalities | | |
|---|---|---|---|---|---|---|---|
| | | Number of samples | Percentage | Cumulative percentage | Number of samples | Percentage | Cumulative percentage |
| eduhigh | 0 | 4762 | 12.62 | 12.62 | 482 | 15.73 | 15.73 |
| | 1 | 8351 | 22.13 | 34.74 | 920 | 30.02 | 45.74 |
| | 2 | 10520 | 27.87 | 62.62 | 807 | 26.33 | 72.07 |
| | 3 | 6962 | 18.45 | 81.06 | 410 | 13.38 | 85.45 |
| | 4 | 3027 | 8.02 | 89.08 | 175 | 5.71 | 91.16 |
| | 5 | 3640 | 9.64 | 98.73 | 241 | 7.86 | 99.02 |
| | 6 | 480 | 1.27 | 100.00 | 30 | 0.98 | 100.00 |
| | All | 37742 | 100.00 | | 3065 | 100.00 | |
| happiness | 1 | 521 | 1.38 | 1.38 | 50 | 1.63 | 1.63 |
| | 2 | 2300 | 6.09 | 7.47 | 222 | 7.24 | 8.87 |
| | 3 | 5193 | 13.76 | 21.23 | 390 | 12.72 | 21.60 |
| | 4 | 22596 | 59.87 | 81.10 | 1832 | 59.77 | 81.37 |
| | 5 | 7132 | 18.90 | 100.00 | 571 | 18.63 | 100.00 |
| | All | 37742 | 100.00 | | 3065 | 100.00 | |

**Table 4. Effects of educational level and ethnic group on residents' subjective well-being.**

| VARIABLES | (1) | (2) |
|---|---|---|
| | happiness | happiness |
| eduhigh | 0.138*** | 0.135*** |
| | (17.51) | (16.75) |
| eduhigh*nation | | 0.042* |
| | | (1.65) |
| nation | | -0.081 |
| | | (-1.21) |
| gender | -0.149*** | -0.149*** |
| | (-7.50) | (-7.48) |
| party | 0.414*** | 0.414*** |
| | (13.23) | (13.21) |
| status14 | 0.179*** | 0.180*** |
| | (15.12) | (15.13) |
| child | 0.131*** | 0.131*** |
| | (13.13) | (13.15) |
| marriage | 0.287*** | 0.287*** |
| | (11.58) | (11.59) |
| observations | 40,431 | 40,431 |
| year | YES | YES |

Note: () inside is the z value.

*, ** and *** indicate significance levels of 10%, 5% and 1%, respectively.

## 4. Results and analysis

### 4.1 Effects of educational level on residents' subjective well-being

Table 4 shows that based on the ordered logit model, we can observe the impact of educational level on residents' subjective well-being. It can be seen from Table 4 that in Column (1), the coefficient of residents' educational level is significantly positive at the level of 1%.

To test Hypothesis 2 and examine the moderating effect of ethnic factors on the relationship between education level and residents' subjective well-being, this paper introduces an interaction term, "eduhigh*nation." The regression results are presented in Table 4 (Column 2). At the 10% significance level, the coefficient of the interaction term (eduhigh*nation), which captures the combined effect of ethnic background and education level, is significantly positive. This finding supports Hypothesis 2, indicating that ethnic factors strengthen the positive correlation between education level and subjective well-being.

The results reveal that the impact of education on the subjective well-being of ethnic minority residents differs from that of Han residents. Specifically, the improvement in education level has a more pronounced effect on enhancing the subjective well-being of ethnic minority residents compared to Han residents. This may stem from the positive impact of education in increasing social mobility and expanding access to resources. In addition, raising the level of education of ethnic minority residents helps to break down stereotypes associated with ethnic identity and social class, reduce barriers between ethnic groups, and enhance psychological belonging and social inclusion. Education can also improve ethnic minority residents' ability to access public services, development opportunities and welfare resources, thus significantly improving their subjective well-being.

## 4.2 Robustness test and endogeneity treatment

Endogeneity is a common issue in empirical analysis, often arising from measurement errors, reverse causality, and omitted variables. For instance, previous studies have demonstrated that happiness significantly influences residents' employment decisions and fertility intentions. Moreover, factors such as class self-positioning, family economic status, and happiness are inherently subjective evaluations. Additionally, as a reflection of subjective cognition, happiness may, in turn, impact respondents' life and work, subsequently influencing their self-assessment of social class. This suggests the possibility of reverse causality between happiness and self-evaluation of class. While the present analysis does not explore endogeneity in detail, this study attempts to mitigate its effects by controlling for variables relevant to the analysis. However, the models may still be subject to bias caused by omitted variables, which remains a limitation of this research.

In order to deal with the endogeneity problem caused by known and unknown factors, reduce the adverse impact of endogeneity on model estimation, and ensure the reliability and robustness of regression results, this study tries to use the following methods for testing. First, we change the estimation method of the explanatory variable, the educational level of residents, and adopt the years of education of residents. According to the CGSS questionnaire "What is your current highest level of education? No education was assigned 0; private school, literacy class and primary school were assigned 6; junior high school was assigned 9; vocational high school, general high school, technical secondary school and technical school were assigned 12; junior college (adult higher education) and junior college (formal higher education) were assigned 15; Undergraduate (adult higher education) and undergraduate (formal higher education) are assigned a value of 16, and graduate students and above are assigned a value of 19, according to which the years of education variable is obtained. Second, in order to avoid the unrobust results caused by the selection of regression methods, this paper further uses the ordered probit model and OLS regression with robust standard errors to re-test the model. Thirdly, this paper tries to adopt the instrumental variable method and selects the educational level of the resident's father (edu_fa) as the instrumental variable of his own educational level [43]. The reasons are as follows: on the one hand, the "elite cycle theory" school of education believes that if the parents are highly educated, their children will have more innate advantages in education, that is, the education level of residents is related to the education level of their parents; On the other hand, the educational level of parents is an exogenous variable in the time dimension that precedes the subjective well-being of residents, the explained variable, and is uncorrelated with the random disturbance term.

In addition, we included the Negative Emotional Impact Scale (PANAS) to provide a more comprehensive and nuanced structural assessment that would enhance the validity of the findings. Specifically, we referenced the question from the CGSS questionnaire: "How many times have you felt depressed or depressed in the past four weeks?" "Always" is assigned a value of 5, "often" is assigned a value of 4, "sometimes" is assigned a value of 3, "rarely" is assigned a value of 2, and "never" is assigned a value of 1. Based on this, we generated the "depressed" variable.

The regression results are shown in Table 5, and it can be observed from Column (1) of Table 5 that the coefficient of the educational years is 0.043, significantly positive at the level of 1%. It can be observed from columns (2) and (3) of Table 5 that the coefficients of explanatory variables are significantly positive at the level of 1%. After changing the model regression method, the regression results are the same as the ordered probit regression results, and education significantly improves the subjective well-being of residents. The combination of multiple methods shows that the main results of this paper are relatively robust. As can be seen from column (4) of Table 5, the coefficient of educational attainment of the explanatory variable is

**Table 5. Robustness test and endogeneity treatment.**

| VARIABLES | (1) | (2) | (3) | (4) |
|---|---|---|---|---|
| | ologit | oprobit | ols | ologit |
| | happiness | happiness | happiness | depressed |
| eduyear | 0.043*** | | | |
| | (15.34) | | | |
| eduhigh | | 0.080*** | 0.064*** | -0.147*** |
| | | (17.95) | (19.75) | (-20.02) |
| gender | -0.160*** | -0.081*** | -0.055*** | -0.174*** |
| | (-8.04) | (-7.27) | (-6.73) | (-9.41) |
| party | 0.466*** | 0.239*** | 0.158*** | -0.286*** |
| | (15.19) | (13.20) | (13.24) | (-8.94) |
| status14 | 0.188*** | 0.100*** | 0.074*** | -0.153*** |
| | (15.90) | (14.99) | (15.44) | (-13.55) |
| child | 0.120*** | 0.068*** | 0.046*** | 0.079*** |
| | (12.22) | (12.34) | (11.19) | (9.17) |
| marriage | 0.251*** | 0.162*** | 0.135*** | -0.263*** |
| | (10.13) | (11.89) | (12.96) | (-11.79) |
| constant | | | 3.423*** | -0.147*** |
| | | | (184.21) | (-20.02) |
| observations | 40,431 | 40,431 | 40,418 | 40,353 |
| year | YES | YES | YES | YES |

Note: () inside is the z value.

*, ** and *** indicate significance levels of 10%, 5% and 1%, respectively.

negative at the significance level of 1%. The results suggest that increasing the level of education can significantly reduce the frequency of depression. The positive effect of education on improving residents' subjective well-being is verified from the side.

Table 6 reports the estimation results using the two-stage least squares method (2SLS), and the first and third columns are the first-and second-stage regression results respectively. The results of the first stage show that the increase of the father's education level can significantly improve the individual education level of residents at the level of 1%, indicating that the education level of residents themselves is highly correlated with the education level of their parents. In addition, the F-statistic of the correlation test between instrumental variables and

**Table 6. Regression results of instrumental variables.**

| VARIABLES | first stage | | second stage | |
|---|---|---|---|---|
| | coefficient | T | coefficient | Z |
| edu_fa | 0.139*** | 92.13 | | |
| eduhigh | | | 0.102*** | 13.40 |
| gender | 0.207*** | 17.82 | -0.059*** | -6.91 |
| party | 1.011*** | 49.55 | 0.106*** | 7.21 |
| status14 | 0.191*** | 28.35 | 0.062*** | 11.22 |
| child | -0.340 *** | -64.41 | 0.064 *** | 11.48 |
| marriage | -0.046 *** | 8.38 | 0.140 *** | 13.07 |
| constant | 1.531*** | -3.24 | 3.334*** | 136.15 |
| Year | Control | | Control | |
| observations | 37563 | | 37563 | |

explanatory variables is significantly greater than 10, indicating that there is no problem of weak instrumental variables, which further confirms that the instrumental variables selected in this paper meet the correlation hypothesis. In addition, the P value of the exogeneity test of the instrumental variable is 0.2, indicating that the educational level of the instrumental variable parents is exogenous and has nothing to do with the disturbance term. It can be seen that the instrumental variables selected in this paper are valid. The second-stage regression results show that residents' educational level will still positively and significantly affect residents' subjective well-being after using instrumental variables, which is basically consistent with the benchmark regression results of this paper.

## 5. Additional tests

The above results show that improving the educational level of residents can significantly improve individual subjective well-being. Ethnic factors positively moderates the relationship between education level and subjective well-being of residents, that is, in the minority residents, the effect of improving education level on subjective well-being of residents is more significant. However, how improving residents' educational level improves subjective well-being and its mechanism need further discussion. In addition, is there a certain difference in the action mechanism between Han and minority residents? This section will explain and explore the reasons why the educational level of ethnic minority residents has a more significant impact on residents' subjective well-being from three perspectives: economy, status, fairness.

### 5.1 Economy

Education exhibits both consumption attributes and productive effects [45]. The consumption attribute refers to the physical and mental happiness education provides, offering individuals spiritual satisfaction. In contrast, the productive attribute highlights how education enhances individuals' earning potential and future development, enabling them to build human capital, improve economic conditions, and ultimately influence their subjective well-being. Specifically, in the context of asymmetric information in the labor market, while educational attainment may not fully represent an employee's capabilities, it serves as a signal to employers, indicating high potential [46]. Consequently, individuals with higher educational levels are more likely to secure desirable jobs and achieve higher income returns. Powdthavee et al. (2015) found that higher levels of education are associated with greater economic benefits in the labor market, which in turn increase happiness and satisfaction [13]. As educational attainment improves, individuals gain enhanced income-generating capabilities, further boosting their subjective well-being. Thus, higher education equips individuals with greater knowledge and skills, enabling them to secure higher wages, improve their quality of life, and enhance their internal sense of happiness and satisfaction [47]. Income can be categorized into "absolute income" and "relative income." The absolute income effect suggests that higher education leads to higher income, which directly contributes to greater happiness. Conversely, the relative income effect posits that individual happiness arises not from an absolute increase in income but from comparisons that determine relative income levels. This study examines the impact of educational attainment on residents' relative income, how it influences subjective well-being, and whether these mechanisms differ between Han and minority ethnic groups.

From Table 7, Column (2), it is evident that with relative income as the dependent variable, the coefficient of educational level is 0.245, which is significantly positive at the 1% level. Similarly, Column (3) shows that the regression coefficients for both educational attainment and relative income level are significantly positive at the 1% level. Column (5) further indicates that the coefficient of educational level remains significantly positive at the 1% level. Column

**Table 7. Eduhigh, economy and residents' subjective well-being.**

| VARIABLES | Han | | | Minority nationalities | | |
|---|---|---|---|---|---|---|
| | **(1)** | **(2)** | **(3)** | **(4)** | **(5)** | **(6)** |
| | happiness | economy | happiness | happiness | economy | happiness |
| eduhigh | 0.135*** | 0.245*** | 0.081*** | 0.177*** | 0.241*** | 0.122*** |
| | (16.54) | (29.06) | (9.67) | (5.92) | (7.72) | (3.99) |
| economy | | | 0.703*** | | | 0.695*** |
| | | | (43.42) | | | (12.76) |
| gender | -0.155*** | -0.092*** | -0.140*** | -0.088 | -0.090 | -0.064 |
| | (-7.45) | (-4.47) | (-6.71) | (-1.23) | (-1.27) | (-0.88) |
| party | 0.424*** | 0.568*** | 0.305*** | 0.269** | 0.531*** | 0.150 |
| | (13.12) | (15.62) | (9.31) | (2.13) | (3.75) | (1.18) |
| status14 | 0.185*** | 0.507*** | 0.066*** | 0.121*** | 0.431*** | 0.013 |
| | (14.98) | (37.47) | (5.17) | (2.77) | (9.74) | (0.29) |
| child | 0.134*** | 0.068*** | 0.121*** | 0.109*** | 0.071** | 0.093*** |
| | (12.87) | (6.95) | (11.70) | (3.03) | (2.11) | (2.61) |
| marriage | 0.296*** | 0.352*** | 0.219*** | 0.188** | 0.221*** | 0.149 |
| | (11.51) | (14.23) | (8.53) | (1.99) | (2.59) | (1.58) |
| observations | 37,391 | 37,391 | 37,391 | 3,040 | 3,040 | 3,040 |
| year | YES | YES | YES | YES | YES | YES |

Note: () inside is the t value.

*, ** and *** indicate significance levels of 10%, 5% and 1%, respectively.

(6) demonstrates that improving residents' educational level enhances subjective well-being by increasing their relative income. These results confirm that higher educational attainment significantly raises the relative income levels of both Han and minority residents, thereby improving their subjective well-being.

To further validate these findings, this study applies the Bootstrap method, performing 500 iterations to test the mediating effect of relative income between Han and ethnic minority groups. Table 8 shows that the 95% confidence intervals for relative income in both groups do not include zero, reaffirming that education affects subjective well-being by improving residents' relative income levels. Moreover, the mediating effect size of relative income for the Han ethnic group is 0.0236, accounting for 37.96% of the total effect. In contrast, the mediating effect size for ethnic minority groups is 0.024, accounting for 27.4% of the total effect. These results indicate that while education enhances relative income and subjective well-being for both groups, the magnitude of the mediating effect is more pronounced among Han residents.

## 5.2 Status

The perception of well-being among residents through education can be enhanced not only via material means but also through non-material channels [48]. Humans inherently possess

**Table 8. Mediating effect test results under the Bootstrap method.**

| Effect type | Effect value | Boot standard error | 95% confidence interval | relative effectiveness |
|---|---|---|---|---|
| Mediating effect of economy of Han nationality | 0.0236*** | 0.0009 | [0.0217,0.0255] | 37.96% |
| Mediating effect of economy of Minority nationalities | 0.024*** | 0.040 | [0.0164.0.0323] | 27.40% |

Note: Using the Bootstrap multi-medium test method, set the number of repeated sampling to 500 times.

*, ** and *** represent significance levels of 5%, 1% and 0.1%, respectively.

social attributes, and social stratification is an inevitable outcome of societal development. According to social stratification theory, groups positioned higher in the social hierarchy tend to enjoy greater social prestige, higher income, and more desirable occupations. These advantages provide enhanced access to resources and information, facilitate the expansion of social networks, and promote the accumulation of social capital. The formation of social classes or strata is determined by factors such as class position, occupational status, educational attainment, property ownership, income, power, and social prestige [49]. Education is closely linked to economic conditions and occupational status, serving as the primary vehicle for social mobility [50]. Across all socioeconomic backgrounds, education is regarded as a key pathway to social advancement, particularly among lower strata, where upward mobility through education is often perceived as the only viable route [51]. Nikolaev (2016) further notes that many individuals pursue higher education as a means to improve their social status [32]. This section explores the role of class status in mediating the relationship between educational attainment and subjective well-being, analyzing this dynamic separately for Han and ethnic minority residents.

From Table 9, Column (2), it is evident that with class self-positioning as the dependent variable, the coefficient of educational attainment is significantly positive at the 1% level. Similarly, Column (3) shows that the regression coefficients for both educational attainment and class self-positioning are significantly positive at the 1% level. These findings indicate that for the Han ethnic group, improving educational attainment significantly enhances residents' class status, which in turn improves their subjective well-being. Column (5) further reveals that the coefficient of educational attainment remains significantly positive at the 1% level. Column (6) demonstrates that improving residents' educational level enhances their subjective well-being by elevating their self-perceived class status.

To further validate these results, this section applies the Bootstrap method, performing 500 iterations to test the mediating effect of class self-positioning for both Han and ethnic minority

**Table 9. Eduhigh, status and residents' subjective well-being.**

| VARIABLES | Han | | | Minority nationalities | | |
|---|---|---|---|---|---|---|
| | (1) | (2) | (3) | (4) | (5) | (6) |
| | happiness | status | happiness | happiness | status | happiness |
| eduhigh | 0.135*** | 0.145*** | 0.104*** | 0.177*** | 0.147*** | 0.148*** |
| | (16.54) | (18.25) | (12.48) | (5.92) | (5.01) | (4.90) |
| status | | | 0.643*** | | | 0.617*** |
| | | | (42.67) | | | (11.68) |
| gender | -0.155*** | -0.154*** | -0.124*** | -0.088 | -0.042 | -0.091 |
| | (-7.45) | (-7.71) | (-5.96) | (-1.23) | (-0.61) | (-1.25) |
| party | 0.424*** | 0.469*** | 0.332*** | 0.269** | 0.523*** | 0.145 |
| | (13.12) | (13.66) | (10.14) | (2.13) | (3.89) | (1.14) |
| status14 | 0.185*** | 1.039*** | | 0.121*** | 1.029*** | -0.113** |
| | (14.98) | (66.93) | | (2.77) | (19.11) | (-2.39) |
| child | 0.134*** | 0.130*** | 0.103*** | 0.109*** | 0.140*** | 0.078** |
| | (12.87) | (13.39) | (9.97) | (3.03) | (4.29) | (2.21) |
| marriage | 0.296*** | 0.285*** | 0.241*** | 0.188** | 0.376*** | 0.104 |
| | (11.51) | (11.97) | (9.39) | (1.99) | (4.58) | (1.11) |
| observations | 37,391 | 37,391 | 37,391 | 3,040 | 3,040 | 3,040 |
| year | YES | YES | YES | YES | YES | YES |

Note: () inside is the t value.

*, ** and *** indicate significance levels of 10%, 5% and 1%, respectively.

**Table 10. Mediating effect test results under the Bootstrap method.**

| Effect type | Effect value | Boot standard error | 95% confidence interval | relative effectiveness |
|---|---|---|---|---|
| Mediating effect of status of Han nationality | 0.0144*** | 0.0009 | [0.0126,0.0162] | 23.11% |
| Mediating effect of status of Minority nationalities | 0.150*** | 0.034 | [0.0083,0.0216] | 16.84% |

Note: Using the Bootstrap multi-medium test method, set the number of repeated sampling to 500 times.

*, ** and *** represent significance levels of 5%, 1% and 0.1%, respectively.

groups. Table 10 shows that the 95% confidence intervals for class self-positioning do not include zero for either group, reaffirming that education affects subjective well-being by improving residents' self-perceived class status. Moreover, the results indicate that the mediating effect of class self-positioning is proportionally greater for the Han ethnic group compared to ethnic minority groups.

### 5.3 Fairness

Fairness significantly impacts individuals' social confidence and is a crucial factor influencing perceived subjective well-being [52]. Rawls (1971) emphasized the need to improve the status of the most disadvantaged groups to achieve fairness [53]. Living in a highly unequal environment often triggers perceptions of unfairness, thereby reducing well-being. Oishi and Kesebir (2015) demonstrated that across 34 countries, increasing income inequality disrupts the positive association between economic growth and happiness [54]. When perceptions of social fairness are low, psychological stress tends to increase, negatively affecting subjective well-being. Social fairness is a multidimensional construct that encompasses various components contributing to a fair society [55]. The relationship between education and residents' sense of fairness remains a topic of debate. On one hand, improving educational attainment enhances individual productivity and provides broader opportunities for higher income, increasing individuals' sense of income fairness and fostering a more positive perception of social fairness [56, 57]. On the other hand, individuals with higher education levels often have elevated income expectations, and when these expectations are unmet, it can lead to negative evaluations of social fairness [58]. This duality suggests that education may simultaneously promote and inhibit perceptions of fairness, depending on contextual factors. This section examines the influence of educational attainment on residents' sense of fairness and its subsequent effect on subjective well-being, with separate analyses for Han and ethnic minority groups.

From Table 11, Column (2), with fairness as the dependent variable, the coefficient of educational attainment is −0.033, which is significantly negative at the 1% level. This finding indicates that in the Han ethnic group, higher educational attainment significantly reduces residents' perceptions of fairness. In contrast, Column (3) shows that the regression coefficients for educational attainment and sense of fairness are significantly positive at the 1% level, suggesting that a stronger sense of fairness partially offsets the positive effects of education on subjective well-being. These results indicate that in the Han ethnic group, the sense of fairness acts as a masking factor, inhibiting the positive influence of education on subjective well-being. From Column (5), it is observed that in ethnic minority groups, the coefficient of educational attainment is −0.004, which is negative but statistically insignificant.

To further validate these findings, the study employs the Bootstrap method with 500 iterations to test the mediating effect of the sense of fairness for both Han and ethnic minority groups. As shown in Table 12, the 95% confidence interval for the sense of fairness includes zero in ethnic minority groups, indicating that the sense of fairness does not mediate the

**Table 11. Eduhigh, fairness and residents' subjective well-being.**

| VARIABLES | Han | | | Minority nationalities | | |
|---|---|---|---|---|---|---|
| | (1) | (2) | (3) | (4) | (5) | (6) |
| | happiness | fairness | happiness | happiness | fairness | happiness |
| eduhigh | 0.135*** | -0.033*** | 0.153*** | 0.177*** | -0.004 | 0.187*** |
| | (16.54) | (-4.39) | (18.53) | (5.92) | (-0.16) | (6.24) |
| fairness | | | 0.622*** | | | 0.569*** |
| | | | (51.05) | | | (13.43) |
| gender | -0.155*** | 0.120*** | -0.196*** | -0.088 | 0.115* | -0.138* |
| | (-7.45) | (6.13) | (-9.34) | (-1.23) | (1.67) | (-1.91) |
| party | 0.424*** | 0.330*** | 0.351*** | 0.269** | 0.324*** | 0.181 |
| | (13.12) | (10.12) | (10.66) | (2.13) | (2.87) | (1.38) |
| status14 | 0.185*** | 0.085*** | 0.164*** | 0.121*** | 0.012 | 0.124*** |
| | (14.98) | (7.26) | (13.22) | (2.77) | (0.28) | (2.87) |
| child | 0.134*** | 0.153*** | 0.095*** | 0.109*** | 0.155*** | 0.081** |
| | (12.87) | (15.77) | (9.21) | (3.03) | (4.77) | (2.30) |
| marriage | 0.296*** | -0.166*** | 0.362*** | 0.188** | -0.171** | 0.225** |
| | (11.51) | (-7.02) | (14.02) | (1.99) | (-2.01) | (2.44) |
| observations | 37,391 | 37,235 | 37,235 | 3,040 | 3,022 | 3,022 |
| year | YES | YES | YES | YES | YES | YES |

Note: () inside is the t value.

*, ** and *** indicate significance levels of 10%, 5% and 1%, respectively.

relationship between education and subjective well-being in these groups. Conversely, in the Han ethnic group, the mediating effect size of the sense of fairness is −0.0036, accounting for 5.16% of the total effect. These findings suggest that in the Han ethnic group, education influences perceptions of fairness, and the sense of fairness inhibits the positive impact of education on subjective well-being.

The results in Table 13 indicate that education exerts a significant but differentiated impact on the subjective well-being of Han and ethnic minority groups. Among the Han population, education positively mediates subjective well-being through relative income and class orientation, accounting for 37.96% and 23.11% of the total effect, respectively. However, the influence of education on perceived fairness is negative, accounting for 5.16% of the total effect. This finding suggests that while education enhances Han residents' perceived fairness, it simultaneously weakens its positive impact on subjective well-being.

One possible explanation is that education improves social status and income levels, but it also heightens individuals' sensitivity to social inequality, leading to a stronger sense of relative deprivation and diminished happiness. Additionally, the educated group may hold higher expectations for future income, and when actual income fails to meet these expectations, it

**Table 12. Mediating effect test results under the Bootstrap method.**

| Effect type | Effect value | Boot standard error | 95% confidence interval | relative effectiveness |
|---|---|---|---|---|
| Mediating effect of fairness of Han nationality | -0.0036*** | 0.0011 | [-0.0056,-0.0015] | 5.16% |
| Mediating effect of fairness of Minority nationalities | 0.0167 | 0.0034 | [-0.0049,0.0083] | |

Note: Using the Bootstrap multi-medium test method, set the number of repeated sampling to 500 times.

*, ** and *** represent significance levels of 5%, 1% and 0.1%, respectively.

**Table 13. Mediating effect test results under the Bootstrap method.**

| Effect type | | Effect value | Boots standard error | 95% confidence interval | relative effectiveness |
|---|---|---|---|---|---|
| Han | economy | 0.0236*** | 0.0009 | [0.0217,0.0255] | 37.96% |
| | Status | 0.0144*** | 0.0009 | [0.0126,0.0162] | 23.11% |
| | Fairness | -0.0036*** | 0.0011 | [-0.0056,-0.0015] | 5.16% |
| Minority nationalities | economy | 0.024*** | 0.040 | [0.0164,0.0323] | 27.40% |
| | Status | 0.150*** | 0.034 | [0.0083,0.0216] | 16.84% |
| | Fairness | 0.0167 | 0.0034 | [-0.0049,0.0083] | |

Note: Using the Bootstrap multi-medium test method, set the number of repeated sampling to 500 times.

*, ** and *** represent significance levels of 5%, 1% and 0.1%, respectively.

could result in a negative evaluation of societal well-being. In contrast, among ethnic minority groups, education also has significant positive mediating effects through relative income and class orientation, accounting for 27.40% and 16.84% of the total effect, respectively. However, the mediating effect of perceived fairness is not statistically significant. This suggests that education does not substantially influence the relationship between perceived fairness and well-being in ethnic minority groups. This divergence may be attributed to the relatively disadvantaged position of ethnic minorities in the distribution of social resources. Their sense of fairness is likely shaped more by structural factors than by individual educational attainment. As a result, improvements in perceived fairness driven by education fail to significantly translate into enhanced subjective well-being.

## 6. Discussion

This paper investigates the influence of education on residents' subjective well-being, with a particular focus on the moderating effect of ethnic factors. Our findings align with the established view that education generally has a positive impact on subjective well-being. Specifically, we analyzed the mechanisms through which education affects the subjective well-being of Han Chinese and ethnic minority groups separately. The results indicate that education enhances subjective well-being across all groups, primarily by elevating relative income levels and shaping social class identity, with income serving as a particularly significant mediator. However, among the Han population, education appears to diminish perceptions of fairness, which marginally reduces happiness. This phenomenon was not observed among ethnic minority groups. For ethnic minorities, the improvement in subjective well-being through education may be attributed to increased access to social resources and enhanced social status. Given the critical role of education in augmenting subjective well-being and the differential effects of ethnicity, policymakers should prioritize inclusive educational strategies, particularly for minority groups. These strategies should focus on increasing the availability of educational resources, supporting cultural identity, and enhancing socioeconomic status. Future research should further explore how various types and levels of education influence the subjective well-being of different ethnic groups, with particular attention to the interplay between individual socioeconomic backgrounds and cultural characteristics.

## 7. Limitation

While our study provides valuable insights into the relationship between education, ethnic factors, and subjective well-being, it is not without limitations. First, our analysis primarily focuses on the Han and minority ethnic groups within China. This scope may limit the

generalizability of our findings to other contexts or countries. The unique socio-cultural and political environment in China significantly influences the relationship between education and subjective well-being. Future research should examine these dynamics in other multicultural and multiethnic societies to validate and extend our findings. Second, subjective well-being is a multidimensional construct encompassing emotional well-being, life satisfaction, and psychological well-being. Our study relies on self-reported measures of happiness and satisfaction, which, while valuable, may not fully capture the complexity of subjective well-being. Incorporating more comprehensive measures in future studies could provide a deeper understanding of this construct. Lastly, although our study considers the role of ethnic factors in the relationship between education and subjective well-being, it does not fully explore the potential interactions between education and other individual-level variables, such as gender, age, and socioeconomic status. These factors may significantly influence subjective well-being and represent promising avenues for future research.

## 8. Conclusion

Using data from the China Comprehensive Family Survey (CGSS) for 2015, 2017, 2018, and 2021, this paper empirically examines the impact of education on the subjective well-being of Chinese residents. The key findings are as follows: First, education plays a positive role in enhancing residents' subjective well-being. This conclusion remains robust after addressing endogeneity concerns and conducting a series of robustness tests. Second, the positive impact of education on subjective well-being is more pronounced among ethnic minority groups. Finally, a comparative analysis between Han and ethnic minority groups reveals that education exerts significant but differentiated effects on the subjective well-being of different ethnic groups. Specifically, for both Han and ethnic minority residents, education primarily enhances subjective well-being by increasing relative income levels and shaping social class orientation. However, among the Han population, education negatively affects perceptions of fairness, which partially offsets its positive impact on happiness. This phenomenon may stem from the fact that education raises individuals' social status and income expectations, and unmet expectations can lead to a sense of relative deprivation, thereby reducing happiness. In contrast, among ethnic minority groups, education does not significantly influence the relationship between perceived fairness and happiness. This may be attributed to the unique position of ethnic minorities in the distribution of social resources, which shapes their perceptions of fairness differently.

## Author Contributions

**Data curation:** Jiameng Yuan.

**Funding acquisition:** Kun Zhang.

**Methodology:** Zhenyi Hu, Kun Zhang.

**Resources:** Jiameng Yuan.

**Writing – original draft:** Zhenyi Hu.

**Writing – review & editing:** Yuanyuan Wang.

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
