## [Decision Letter · Decision Letter 0]

3 Oct 2024

PONE-D-24-12668Do higher educated people feel happier?——Evidence of Han and minority nationalities from ChinaPLOS ONE

Dear Dr. Hu,

Thank you for submitting your manuscript to PLOS ONE. After careful consideration, we feel that it has merit but does not fully meet PLOS ONE’s publication criteria as it currently stands. Therefore, we invite you to submit a revised version of the manuscript that addresses the points raised during the review process.

We look forward to receiving your revised manuscript.

Kind regards,

Nik Ahmad Sufian Burhan

Academic Editor

PLOS ONE

Journal Requirements:

“The authors would like to acknowledge the financial support Funded by Science Research Project of Hebei Education Department HB23YJ011 for this research.”

3. Please note that funding information should not appear in the Acknowledgments section or other areas of your manuscript. We will only publish funding information present in the Funding Statement section of the online submission form. Please remove any funding-related text from the manuscript. 

4. For studies involving third-party data, we encourage authors to share any data specific to their analyses that they can legally distribute. PLOS recognizes, however, that authors may be using third-party data they do not have the rights to share. When third-party data cannot be publicly shared, authors must provide all information necessary for interested researchers to apply to gain access to the data. (https://journals.plos.org/plosone/s/data-availability#loc-acceptable-data-access-restrictions) 

Reviewers' comments:

Reviewer's Responses to Questions

**Comments to the Author**

1. Is the manuscript technically sound, and do the data support the conclusions?

Reviewer #1: Yes

Reviewer #2: Yes

2. Has the statistical analysis been performed appropriately and rigorously? 

Reviewer #1: Yes

Reviewer #2: Yes

3. Have the authors made all data underlying the findings in their manuscript fully available?

Reviewer #1: Yes

Reviewer #2: Yes

4. Is the manuscript presented in an intelligible fashion and written in standard English?

Reviewer #1: No

Reviewer #2: Yes

5. Review Comments to the Author

Reviewer #1: Methodological Considerations

Theoretical Framework: The manuscript could benefit from a more robust theoretical framework that explicitly links the research hypotheses to established theories and concepts in the field of well-being and social psychology. For instance, the authors could delve deeper into the theoretical underpinnings of the relationship between education and subjective well-being, drawing on concepts such as self-determination theory, social comparison theory, or the human capital model. Additionally, the authors could elaborate on how cultural values, social norms, and access to opportunities might differ between Han and minority groups, and how these factors could potentially shape the impact of education on well-being.

Measurement of Subjective Well-being: While the use of a single-item measure of happiness is common in some research, it is crucial to acknowledge its limitations in capturing the multifaceted nature of subjective well-being. The authors could consider incorporating additional measures or scales, such as the Satisfaction with Life Scale (SWLS), the Positive and Negative Affect Schedule (PANAS), or other validated measures of psychological well-being. This would provide a more comprehensive and nuanced assessment of the construct and strengthen the validity of the findings.

Causal Inference: The authors have attempted to address endogeneity concerns using instrumental variable analysis, which is commendable. However, it is essential to acknowledge that the cross-sectional nature of the data limits the ability to establish definitive causal relationships. The authors should exercise caution in interpreting their findings as causal effects and consider discussing the potential for reverse causality or the influence of unmeasured confounding variables.

Sample Representativeness: While the CGSS is a valuable resource, the authors should provide more details about its sampling methodology and any potential biases that might affect the representativeness of the sample. This would enable readers to better assess the generalizability of the findings to the broader Chinese population.

Presentation and Clarity

Introduction and Literature Review: The introduction could be more focused and concise, highlighting the key research gap and the specific contribution of the study. The literature review could be more structured and synthesized, clearly identifying the main themes and theoretical perspectives from the existing literature and articulating how the current study builds upon and extends previous research.

Statistical Reporting: The presentation of statistical results could be enhanced by ensuring clarity and consistency. The authors should provide all relevant statistics, including coefficients, standard errors, p-values, and model fit statistics, in a clear and organized manner within the tables.

Language and Style: While the manuscript is generally understandable, the language could be further refined to improve clarity and precision. Some sentences are overly complex or contain grammatical errors, which can hinder comprehension. A thorough proofreading and editing process would significantly enhance the manuscript's overall quality and facilitate effective communication of the research findings.

Specific Examples for Improvement

Theoretical Framework: Instead of simply stating that "education can lead to better material and non-material living conditions," the authors could elaborate on the specific psychological mechanisms through which education might enhance well-being, such as increased self-efficacy, expanded social networks, or greater sense of control over one's life.

Measurement of Subjective Well-being: The authors could consider including a brief discussion of the limitations of single-item measures of happiness and explain why they chose this approach despite these limitations. They could also mention potential future research directions that could employ more comprehensive measures of subjective well-being.

Language and Style: The sentence "Residents' subjective well-being can not only reflect the current situation of individual life, but also potentially act on individual external behavior, and has an important impact on the harmony and stability of the whole society" could be revised to something like: "Subjective well-being reflects an individual's quality of life and can influence their behavior, ultimately contributing to societal well-being."

Reviewer #2: This article provides a comprehensive and insightful analysis of the impact of education on subjective well-being, particularly highlighting the differences between Han and ethnic minority groups in China. The study is well-structured and grounded in a robust dataset from the China General Family Survey (CGSS), contributing to the growing body of research on the relationship between education and well-being. The emphasis on ethnic factors adds a unique and valuable dimension, offering important policy implications for more inclusive educational strategies aimed at improving the socio-economic and well-being outcomes of minority groups.

There are a few minor issues to address.

1. In the Abstract, briefly mention the research methodology (e.g., statistical tools, analysis techniques, sample

size) to give readers insight into the study's rigor.

2. In Model Setting (3.2) sections contain Chinese words that should be changed into English to ensure

consistency and accessibility for an international audience. Explain the justification for using the ordered logit

model.

3. In Section 3.3, please provide the internal consistency reliability of the instrument. In section 3.3.2,

participants were divided into 14 or 13 levels of education. Please check.

4. In Section 4.1, how does Table 4 explain the more significant results of ethnic minority residents compared to

Han?

5. Some typos throughout the article should be corrected for clarity and professionalism. E.g. In Table 9, Column 2

is “status” not “fairness”.

Overall, this paper is a valuable contribution to the field, providing policy-relevant insights and a solid foundation for further exploration of the complex relationship between education, ethnicity, and well-being.

6. PLOS authors have the option to publish the peer review history of their article (what does this mean?). If published, this will include your full peer review and any attached files.

Reviewer #1: No

Reviewer #2: No

---

## [Author Response · Author response to Decision Letter 0]

27 Nov 2024

Dear Editors and Reviewers:

Thank you for your letter and for the reviewers’ comments concerning our manuscript. Those comments are all valuable and very helpful for improving our paper, as well as the important guiding significance to our research. We have read comments carefully and have made revisions which we hope meets with approval. In what follows, we list each comment (in italics) and our response (in roman text). We also highlight the revised content in BLUE in the revised paper.

Response to Reviewer 1: 

1.Theoretical Framework: The manuscript could benefit from a more robust theoretical framework that explicitly links the research hypotheses to established theories and concepts in the field of well-being and social psychology. For instance, the authors could delve deeper into the theoretical underpinnings of the relationship between education and subjective well-being, drawing on concepts such as self-determination theory, social comparison theory, or the human capital model. Additionally, the authors could elaborate on how cultural values, social norms, and access to opportunities might differ between Han and minority groups, and how these factors could potentially shape the impact of education on well-being.

Response: This is a very meaningful suggestion. In order to solve this problem, we thoroughly investigate the relationship between education and subjective well-being by integrating the perspectives of self-determination theory and social comparison. In addition, we will elaborate on the differences between Han and ethnic minorities in cultural values, social norms, and access to opportunities, and how these factors affect the impact of education on well-being.

Position in the paper: See section 2 “Theoretical analysis and research hypothesis ”.

2.Measurement of Subjective Well-being: While the use of a single-item measure of happiness is common in some research, it is crucial to acknowledge its limitations in capturing the multifaceted nature of subjective well-being. The authors could consider incorporating additional measures or scales, such as the Satisfaction with Life Scale (SWLS), the Positive and Negative Affect Schedule (PANAS), or other validated measures of psychological well-being. This would provide a more comprehensive and nuanced assessment of the construct and strengthen the validity of the findings.

Response: Thanks for your suggestions! Thank you for your insightful feedback on the measurement of subjective well-being. We acknowledge the limitations of using a single-item measure of happiness, particularly in capturing the complex and multifaceted nature of subjective well-being. To address this, we will consider incorporating additional validated scales, such as the Negative Affect Schedule (depressed). This provides a view by capturing negative emotional states and allow for a more comprehensive evaluation of subjective well-being, addressing both the cognitive and affective dimensions of the construct. This approach would enhance the robustness of our analysis, provide richer insights into the determinants of well-being, and strengthen the validity and reliability of our findings.

Position in the paper: See section 4.2 “Robustness test and endogeneity treatment”.

3. Causal Inference: The authors have attempted to address endogeneity concerns using instrumental variable analysis, which is commendable. However, it is essential to acknowledge that the cross-sectional nature of the data limits the ability to establish definitive causal relationships. The authors should exercise caution in interpreting their findings as causal effects and consider discussing the potential for reverse causality or the influence of unmeasured confounding variables.

Response: Thank you for this valuable comment. In response, we appreciated the recognition of our efforts to address endogenous issues through instrumental variable analysis. However, we are fully aware of the limitations with our data, and to address this, we discuss potential issues more thoroughly in the robustness test and talk about them in the limitations of this paper, analyze the possibility of unmeasured confounding variables that may affect education and well-being, and emphasize the need for future studies using longitudinal data. To further establish the causal relationship between education and subjective well-being.

Position in the paper: See section 4.2 “Robustness test and endogeneity treatment” and section 7 “Limitation”.

4. Sample Representativeness: While the CGSS is a valuable resource, the authors should provide more details about its sampling methodology and any potential biases that might affect the representativeness of the sample. This would enable readers to better assess the generalizability of the findings to the broader Chinese population.

Response: Thank you for this valuable comment. We recognized the importance of providing a detailed explanation of the sampling methodology used in the China General Social Survey (CGSS) to ensure transparency and to help readers assess the generalizability of our findings. In response to this, we included a more comprehensive description of the CGSS sampling process in the manuscript. The CGSS employs a multi-stage, stratified, and probabilistic sampling method that aims to capture a nationally representative sample of the Chinese adult population. This process involves selecting respondents across different provinces, cities, and rural areas, ensuring that the sample reflects the geographic, demographic, and socio-economic diversity of China.

Position in the paper: See section 3.1 “Sample selection”.

5.Presentation and Clarity Introduction and Literature Review: The introduction could be more focused and concise, highlighting the key research gap and the specific contribution of the study. The literature review could be more structured and synthesized, clearly identifying the main themes and theoretical perspectives from the existing literature and articulating how the current study builds upon and extends previous research.

Statistical Reporting: The presentation of statistical results could be enhanced by ensuring clarity and consistency. The authors should provide all relevant statistics, including coefficients, standard errors, p-values, and model fit statistics, in a clear and organized manner within the tables.

Language and Style: While the manuscript is generally understandable, the language could be further refined to improve clarity and precision. Some sentences are overly complex or contain grammatical errors, which can hinder comprehension. A thorough proofreading and editing process would significantly enhance the manuscript's overall quality and facilitate effective communication of the research findings.

Response: Thank you for the suggestion. We appreciated your constructive suggestions, which will help us improve the overall quality and readability of our study. We refined the introduction. We restructured the literature review to provide a more synthesized overview of the main themes and theoretical perspectives, such as Self-Determination Theory, Social Comparison Theory, and Human Capital Theory. This would help clarify how our research builds upon and extends previous studies. To enhance the presentation of our statistical results, we ensured that all relevant statistics are reported in a clear and consistent manner. We conducted a thorough proofreading and editing process to simplify overly complex sentences, correct grammatical errors, and enhance the overall readability. 

Response to Reviewer 2: 

1.In the Abstract, briefly mention the research methodology (e.g., statistical tools, analysis techniques, samplesize) to give readers insight into the study's rigor.

Response: Thank you for the valuable suggestion. I have included a brief mention of the research methodology in the abstract, covering the statistical tools, analysis techniques, and sample size, to provide readers with better insight into the study's rigor.

Position in the paper: See section “Abstract”.

2.In Model Setting (3.2) sections contain Chinese words that should be changed into English to ensure consistency and accessibility for an international audience. Explain the justification for using the ordered logit model.

Response: Thank you for pointing out this issue. I have replaced the Chinese terms in Section 3.2 (Model Setting) with English to enhance the paper's consistency and accessibility for an international audience. Additionally, I have clarified the rationale for using the ordered logit model. Given that the two discrete variables, class self-positioning and happiness, have an inherent ordinal structure, the ordered logit model is well-suited for this analysis. This model allowed us to capture the ordered nature of the data while providing reliable estimates of how the independent variables influence the probability of higher or lower outcome levels. This alignment with the data's characteristics made the ordered logit model an appropriate and effective choice for our study.

Position in the paper: See section 3.2 “Model setting”.

3.In Section 3.3, please provide the internal consistency reliability of the instrument. In section 3.3.2, participants were divided into 14 or 13 levels of education. Please check.

Response: Thank you for your attention to these details. I have provided the internal consistency reliability of the instrument to ensure transparency regarding the measure's reliability and carefully verified the educational levels and clarify whether participants were divided into 13 or 14 levels to ensure accuracy in reporting.

Position in the paper: See section 3.3.2 “Educational level (eduhigh)”.

4. In Section 4.1, how does Table 4 explain the more significant results of ethnic minority residents compared to Han?

Response: Thank you for your question. In Section 4.1, column (2) of Table 4 contained the interaction between nation and the education level of residents. The results showed that the coefficient of eduhigh*nation is significantly positive at the level of 10%, and ethnic background positively regulates the relationship between education level and subjective well-being. This finding suggested that the impact of education on the subjective well-being of ethnic minority residents is different from that of Han residents. The stronger effect of education on the subjective well-being of ethnic minority residents may mean that education plays a relatively larger role in improving their subjective well-being, possibly due to factors such as increased social mobility or access to resources.

Position in the paper: See section 4.1 “Effects of educational level on residents' subjective well-being”.

5. Some typos throughout the article should be corrected for clarity and professionalism. E.g. In Table 9, Column 2 is “status” not “fairness”.

Response: Thank you for pointing out these issues. I have carefully reviewed the article to correct any typographical errors and ensure clarity and professionalism throughout.

Position in the paper: See section 5.2 “Status”.

We tried our best to improve the manuscript and made some minor changes in the manuscript. These changes will not influence the content and framework of the paper. We appreciate for Editors/Reviewers’ warm work earnestly, and hope that the correction will meet with approval.

Once again, thank you very much for your comments and suggestions.

---

## [Editor Report · Decision Letter 1]

9 Dec 2024

Do higher educated people feel happier?

——Evidence of Han and minority nationalities from China

PONE-D-24-12668R1

Dear Dr. Hu,

We’re pleased to inform you that your manuscript has been judged scientifically suitable for publication and will be formally accepted for publication once it meets all outstanding technical requirements.

Kind regards,

Nik Ahmad Sufian Burhan

Academic Editor

PLOS ONE
---

## [Editor Report · Acceptance letter]

2 Jan 2025

PONE-D-24-12668R1 

PLOS ONE

Dear Dr. Hu, 

I'm pleased to inform you that your manuscript has been deemed suitable for publication in PLOS ONE. Congratulations! Your manuscript is now being handed over to our production team.

Kind regards, 

on behalf of

Dr. Nik Ahmad Sufian Burhan 

Academic Editor

PLOS ONE